# Danish dog owners' use and the perceived effect of unlicensed cannabis products in dogs

**Pernille Holst**[ID]*, **Annemarie Thuri Kristensen, Maja Louise Arendt**[ID]*

Department of Veterinary Clinical Sciences, Faculty of Health and Medical Sciences, University of Copenhagen, Copenhagen, Denmark

* ph@sund.ku.dk (PH); maja.arendt@sund.ku.dk (MLA)

**Data Availability Statement:** All relevant data are within the manuscript and its Supporting information files.

**Funding:** The author(s) received no specific funding for this work.

## Abstract

The interest in the use of medical cannabis has increased in recent years in both human and veterinary fields. In Denmark, there are no veterinary-licensed medical cannabis or cannabinoid supplements, and it is illegal to prescribe or sell cannabinoids intended for the treatment of veterinary patients. This study aimed to explore the unlicensed cannabinoid use in Danish dogs, by questioning dog owners about usage, indication for use, way of purchase, and their perceived effect of the cannabinoid treatment. An anonymous online survey was distributed via social media. The total number of respondents were 2,002, of which 38% indicated using or having administered cannabinoids to their dog. The majority of the respondents confirming the use of cannabinoids (93%) had used cannabidiol drops/oil and only few (4%) reported using Δ9-tetrahydrocannabinol-based products. Most owners (67%) purchased the products online. The three most common indications for use were pain alleviation, behavioural issues, and allergy. When asked about the respondent-perceived effect the majority reported a good or very good effect. The indication with the highest percentage of owner-perceived positive effect (77%) was pain alleviation. This study shows that, despite no licensed veterinary cannabinoid products being available in Denmark, dog owners do supplement their dogs with cannabinoids and the majority of these perceive that the treatment had a positive effect. This supports the need for more evidence-based knowledge in veterinary cannabinoid therapy.

## Introduction

In recent years, the use of medical cannabis has received renewed interest in both human and veterinary medicine. After the discovery of the endocannabinoid system in the late 1980s [1], researchers have been exploring the system as a potential treatment target [2]. Cannabis is the popular name for the plant Cannabis sativa and products derived from the plant. Cannabis sativa contains over 100 cannabinoids with cannabidiol (CBD) and Δ9-tetrahydrocannabinol (THC) being the two main cannabinoids [3]. In human medicine, a substantial amount of clinical research concerning medical cannabis has been conducted and is currently ongoing. A search in the WHO International Clinical Trials Registry Platform in September 2022, showed

**Competing interests:** The authors have declared that no competing interests exist.

265 active clinical trials registered [4]. Several pre-clinical and pharmacokinetic studies on veterinary cannabinoids have been conducted [5–18] but only a limited number of clinical trials in dogs have been published. The majority of studies have focused on investigating the effects of cannabinoids on osteoarthritis-related pain, but there have also been reports of studies on idiopathic epilepsy, behavioural modification, and canine atopic dermatitis [16, 19–31].

There is an increasing availability of commercial cannabis-based extracts, nutraceuticals, supplements, feed additives, and treats aimed at companion animals. Unlike medical products, these types of cannabis-based products are not subject to regulation, including standardized evaluation of cannabinoid quality and quantity, screening for contaminants, and microbial testing. In a market review by Bonn-Miller et al. (2017), the content and labelling of 84 commercially available CBD extracts on the US market were analyzed. Of these, only 32% correctly labeled the actual CBD content, with 42% of products being under-labelled, and 26% being over-labelled. In addition, non-labelled cannabinoids, including THC, were found in up to 20% of the products [32]. Similar results have been reported in commercially available CBD extracts in the European Union, with some products containing potentially hazardous contaminants and most lacking information about THC content [33]. A study by Wakshlag et al. (2020) analyzed the content of THC, CBD, other selected cannabinoids, terpenes, and heavy metals in 29 full-spectrum commercially available veterinary products. They found that only 10 of the 29 products fell within 10% of the label claims regarding cannabinoid content, and 4 of the 29 products contained heavy metals [34]. When dog owners self-prescribe these unregulated CBD products, the lack of consistency in labelling content poses a significant challenge to accurately determining dosage and assessing effects. In the worst-case scenario, this inconsistency may pose a health risk to the dog, particularly concerning THC and heavy metal intoxication. In Denmark recreational cannabis is prohibited, and medical cannabis has been illegal up until 2018 when a national pilot programme was passed by the Danish government, allowing human physicians to prescribe medical cannabis for certain indications. The purpose of the programme was to enable the prescription of medical cannabis to patients experiencing a lack of effect from conventional therapy. Veterinary prescription is not included in the programme [35]. In Denmark, there are no approved veterinary medical cannabis products, and no veterinary cannabinoid supplements or feed additives are registered by the Danish Veterinary and Food Administration, Ministry of Food, Agriculture and Fisheries. As a result, it is illegal to sell or prescribe cannabinoids for the treatment of veterinary patients. There are many anecdotal reports on the use and effect of cannabinoid products for various health conditions in companion animals. Three surveys have investigated the use of veterinary cannabinoids among companion animal owners in the US and Canada. The results showed that the most frequent reasons for use were managing seizures and pain, and there was a generally positive perception of its effectiveness by the owners [36–38].

This study aimed to investigate the use of unauthorized cannabinoids in Danish dogs, including indications for use as well as the owner-perceived effect. An underlying incentive was furthermore to provide information on indications with high owner-perceived effects, which future academic research could build upon. Given the lack of legal veterinary cannabinoid products, an anonymous survey was deemed the most appropriate method to obtain this information.

## Material and methods

### Survey

The questionnaire was developed, and the data was collected using the survey platform SurveyXact by Ramboll, (Rambøll, Denmark). A pilot questionnaire was designed and reviewed

by the co-investigators and was tested by staff at the University of Copenhagen, Denmark. Feedback was provided in an iterative process and the questionnaire was refined to eliminate inappropriate branching, improve flow, and correct misleading response options. The study was approved by the Ethical and Administrative Committee, Department of Veterinary Clinical Sciences, University of Copenhagen (study identification number EAU:2020–20).

Participants were selected by convenience sampling. The open survey was advertised via the Facebook page of the University Hospital for Companion Animals, University of Copenhagen. In addition, various dog societies and breeder associations were contacted, and they shared the survey link via their social media platforms.

An introduction with the aim of the study, the number of questions, the expected time for completion, and a declaration of anonymity were provided at the beginning of the questionnaire. The survey consisted of a maximum of eight structured questions in a branching series with only relevant questions being displayed based on the respondent's reply to previous questions. Upon completion of the questionnaire, the respondents agreed that the data could be used for research and publication. The full questionnaire can be viewed in S1 Appendix. All data were collected anonymously, and no personal data were collected or stored. The online questionnaire was available from July 2019 to November 2019.

**Demographic variables.** The first two questions included demographic variables including which area in Denmark the respondents were located (greater Copenhagen, northern Zealand, other Zealand, Funen, northern Jutland, central Jutland, southern Jutland, islands) and what their dog weighed in kilograms (small < 10 kg, medium 10–20 kg, large 20–30 kg, extra-large/giant >30 kg).

**Cannabinoid use, product type, indication and way of purchase.** The respondents were questioned about their use or past use of cannabinoid products for their dogs. Those who confirmed usage were directed to a subsequent section containing more detailed inquiries and were asked (a) what type of product/s they had used (CBD drops/oil, CBD capsules, CBD ointment/cream, CBD powder, CBD spray, products primarily containing THC, cannabis tea, other formulations of cannabis/hemp products). The last option had an open-ended text box where the respondents could add-in additional information on cannabinoid formulations/types not covered by the questionnaire. Respondents could select more than one indication; (b) for which indications they used cannabinoids (treatment of cancer, treatment of pain, treatment of poor appetite, treatment of gastrointestinal disease or clinical signs, disease prevention or well-being purposes, treatment of allergy, treatment of seizures). Additionally, there was an open-ended text box for alternative indications. Respondents could select more than one indication; (c) if they experienced an effect of the treatment (very convincing and good effect, convincing with some effect, possible but not convincing effect, no effect); (d) where they purchased the cannabinoid product (internet, pharmacy, herbalist, abroad, at a market). For alternative or elaborate answers an open-ended text box was available. All open-ended text boxes were checked for answers which should have been included in the predefined categories and the data was adjusted so they were included in the appropriate predefined categories.

**Use of other herbal remedies.** All respondents, regardless of their response to the use of cannabinoids, were asked if they had treated their dogs with other herbal remedies.

## Statistical analyses

Descriptive and simple percentage data were collected via SurveyXact by Ramboll (Rambøll, Denmark) and figures were generated in GraphPad Prism version 8.00 for Windows (Graph-Pad Software, La Jolla, California, USA). The survey data were downloaded into IBM SPSS 26

(IBM, New York, USA) for statistical analysis. As not all respondents replied to all questions and some questions had the option of multiple answers, there is a difference in total sums between questions. The percentages were calculated based on the total number of responses to the individual questions. The association between the categorical variables "geographic location" and "the choice to treat with cannabinoids" was assessed using a chi-square test. Binomial logistic regression analysis was performed with 95% confidence intervals to determine the odds ratios for cannabinoid use in the geographic regions. The statistical analysis for indication and owner-perceived effect of treatment was performed using the non-parametric Kruskal-Wallis test followed by a post-hoc test using Dunn's test applying Bonferroni correction for multiple testing. For all statistical analyses, a $P$ value less than 0.05 was considered statistically significant. The inclusion criteria for participation were dog owners in Denmark, and all non-completed questionnaires were excluded from the analysis.

## Results

Out of 2,091 respondents who initiated the survey, 2,002 respondents completed the survey for a completion rate of 95.7%. Only completed surveys were included in the analyses.

### Demographics

The crude demographics showed respondents were concentrated around the two largest cities in Denmark (capital region, n = 1,044, 52% and central Jutland, n = 343, 17%). The weight of the respondents' dogs was distributed as 24% (n = 486) being under 10 kg, 24% (n = 470) between 10 and 20 kg, 25% (n = 501) between 20 and 30 kg, and 27% (n = 545) over 30 kg (Table 1).

### Use of cannabinoid products for dogs

Of the 2,002 completed questionnaires, 752 (38%) respondents reported having used at least one cannabinoid product for their dog, while 1,250 (62%) had never used such products. A

**Table 1. Demographic characteristics of the respondents.**

| Demographic characteristics | Respondents | |
|---|---|---|
| | n | % |
| Completed questionaires | 2,002 | 96 |
| Partial completed questionaires | 89 | 4 |
| Location | | |
| Greater Copenhagen (Capital region) | 430 | 22 |
| Northern Zealand (Capital region) | 209 | 10 |
| Other Zealand (Capital region) | 405 | 20 |
| Funen | 190 | 10 |
| Northern Jutland | 186 | 9 |
| Central Jutland | 343 | 17 |
| Southern Jutland | 188 | 9 |
| Islands | 51 | 3 |
| Dog weight | | |
| Small < 10 kg | 486 | 24 |
| Medium 10–20 kg | 470 | 24 |
| Large 20–30 kg | 501 | 25 |
| Extra-large or giant >30 kg | 545 | 27 |

chi-square test was used to assess the association of living area, with the reporting of treatment with cannabinoids, and a significant association was found ($X^2$ (7, N = 2,002) = 16.61, $p$ = 0.020). Binominal logistic regression revealed that living in greater Copenhagen was negatively associated with cannabinoid use in dogs, with the respondents having an odds ratio of 0.69 related to using cannabinoids (P = 0.04, OR 0.69, 95% CI = 0.48–0.98).

## Cannabinoid formulation and type

Out of the 752 respondents who reported having treated their dog with at least one cannabinoid product, the most commonly used product was CBD drops/oil, used by 697 (93%) respondents. CBD ointment/cream was the second most commonly used product, used by 67 (9%) respondents, while CBD capsules and spray were used by 29 (4%) and 27 (4%) respondents respectively. CBD powder was used by seven (1%) respondents. Thirty-six (5%) respondents indicated having used other cannabinoid or hemp formulations and 33 (4%) reported experience with the use of products primarily containing THC. Of all the respondents confirming the use of cannabinoids, 113 (15%) had used more than one cannabinoid formulation or type, with the majority having tried two different formulations or types. Three respondents indicated having tried four different cannabinoid formulations or types. The evaluation of 36 open-ended text box answers showed that 15 of the responses should have been included in the predefined categories. When adding these (six CBD oil, two CBD cream, and seven CBD capsules) to the appropriate categories, the overall frequencies were not affected. The open-ended text boxes also revealed additional cannabinoid formulations and types not included in the predefined categories. The most common were hemp oil (n = 12) and hemp seed oil (n = 4), followed by homemade mixtures (n = 2) and hemp tablets (n = 1). There were also two unusual responses, one related to 'secondary marijuana smoke inhalation' and another to 'marijuana mixed with food'.

## The indication for the use of cannabinoids

Pain management was the most common reason for using cannabinoids, with 36% (n = 376) of respondents reporting this as a reason. The second most common indication was allergy (11%, n = 122), followed by disease prevention/well-being (9%, n = 100) and seizure control (4%, n = 40). Of the inappropriate free text responses nine respondents recorded "epilepsy" in the free text and not in the predefined category "seizure control". Similarly, eight respondents recorded "well-being" as free text, instead of indicating "disease prevention or well-being purposes" in the predefined category. Two additional major categories of indication were identified from the 28% (n = 299) of respondents who typed in other indications as free text. These were "behavioural issues" (n = 154), and "dermatological disease" (n = 34). Overall, the behavioural issue category was the second-largest indication group and included conditions such as anxiety, stress, fireworks-, and thunder phobia. The remaining indications were miscellaneous and included: neurological disease including dementia, intervertebral disc disease, and meningitis (n = 21); respiratory disease including pneumonia, cough, and excess sputum production (n = 12); cancer including treatment of splenic tumor, of an undiagnosed tumor, and cancer prevention (n = 3), palliative purposes (n = 5); reproductive issues including stimulation of neonatal weight gain, stabilizing hormonal status in a pregnant bitch, and mastitis (n = 2); immunological disease including autoimmune disease (n = 2); as an anti-inflammatory agent (n = 2); renal disease (n = 2); pancreatitis (n = 1); heart disease (n = 1); prostatic disease (n = 1); as bacteriostatic agent (n = 1); immune-stimulation (n = 1), and prevention of medical side effects from other treatments (n = 1). The corrected distribution of indications for

**Table 2. The distribution of the indications for which owners administered cannabinoids to their dogs.**

| Indications[b] | Responses | |
|---|---|---|
| | n | % |
| Pain | 403 | 39 |
| Behavioural issue | 154 | 15 |
| Allergy | 122 | 12 |
| Disease prevention or well-being | 108 | 10 |
| Cancer | 71 | 7 |
| Seizure | 49 | 5 |
| Gastrointestinal disease | 35 | 3 |
| Dermatological disease | 34 | 3 |
| Poor appetite | 19 | 2 |
| Other | 42 | 4 |
| Total | 1,037[a] | 100 |

[a] The respondents (n = 752) could choose more than one indication for cannabinoid use, explaining the total number of indications being 1,037.

[b] Corrected and adjusted for inappropriate free-text registrations.

cannabinoid use is reflected in Table 2, which considers the adjustments made to the categorical sums.

## Owner-perceived efficacy of cannabinoid treatment

Out of the 752 respondents who had used cannabinoids for their dogs, 48% (n = 363) reported a "very convincing and good effect", 29% (n = 217) indicated "convincing with some effect", whilst 18% (n = 137) were uncertain of the effect ("possible, but not convincing effect"), and only 5% (n = 35) reported observing "no effect".

**Difference in owner-perceived efficacy between indication groups.** When analyzing the indication-specific owner-perceived effect of the cannabinoid treatment, only respondents reporting one treatment indication were included, as it was not possible to distinguish the effect between various indications. Of the 752 respondents confirming use of cannabinoids, 69% (n = 516) used cannabinoids for only one indication and the majority reported some degree of effect. For the three largest indication groups, pain management (n = 249), behavioural issue (n = 103), and allergy (n = 47), the perceived overall effect was high (Table 3).

**Table 3. The owner-perceived effect of cannabinoid treatment in different indication groups.**

| Owner-perceived effect | | Pain | Behavioural issue | Allergy | Cancer | Seizure | Well-being | Gastro-intestinal disease | Dermato-logical disease | Other | Appetite | Total |
|---|---|---|---|---|---|---|---|---|---|---|---|---|
| Very convincing and good effect | n | 113 | 39 | 14 | 19 | 16 | 10 | 8 | 4 | 4 | 1 | 228 |
| | % | 45 | 38 | 30 | 54 | 47 | 40 | 80 | 67 | 67 | 100 | 44 |
| Convincing with some effect | n | 80 | 30 | 13 | 6 | 4 | 5 | 1 | 2 | 2 | 0 | 143 |
| | % | 32 | 29 | 28 | 17 | 12 | 20 | 10 | 33 | 33 | 0 | 28 |
| Possible, but no convincing effect | n | 43 | 25 | 16 | 8 | 10 | 10 | 1 | 0 | 0 | 0 | 113 |
| | % | 17 | 24 | 34 | 23 | 29 | 40 | 10 | 0 | 0 | 0 | 22 |
| No effect | n | 13 | 9 | 4 | 2 | 4 | 0 | 0 | 0 | 0 | 0 | 32 |
| | % | 5 | 9 | 9 | 6 | 12 | 0 | 0 | 0 | 0 | 0 | 6 |
| Total | n | 249 | 103 | 47 | 35 | 34 | 25 | 10 | 6 | 6 | 1 | 516 |

The respondent group using cannabinoids for pain management reported the highest percentage of perceived effect. Seventy-seven percent (n = 193) indicated either "very convincing and good effect" or "convincing with some effect". Only five percent (n = 13) reported "no effect", and 17% (n = 43) were unsure of the effect ("possible, but not convincing effect"). For behavioural issues, the majority (67%, n = 69) observed an effect. Nine percent (n = 9) reported "no effect", and 24% (n = 25) reported a "possible, but not convincing effect". When used for treatment of allergy, 57% (n = 27) reported a positive effect with an equal distribution of "very convincing and good effect" (30%, n = 14) and "convincing with some effect" (28%, n = 13). When the categories for effect in the allergy group were analyzed individually, the largest number of respondents were in the category "possible, but not convincing effect" with 34% (n = 16) indicating a lack of obvious effect. Nine percent (n = 4) reported "no effect". The remaining indication groups had low total respondent numbers (n = 1–25). The indication that had the highest percentage of owners not perceiving any effect (12%, n = 4) was when used for seizure control (n = 34). As differences in the owner-perceived effect between the indication groups could reflect a true difference in response to treatment of different disease states a Kruskal-Wallis test was performed and provided weak evidence of a difference $\chi^2(9, N = 516) = 19.808$, $p < 0.019$, $E^2 = 0.40$. A multiple comparison post hoc test with a Dunn's test did however not show any significant differences in perceived effect between the indication groups ($p > 0.05$, adjusted using the Bonferroni correction).

## Purchase

The majority of respondents (67%, n = 507) purchased the cannabinoid product online, 6% (n = 42) purchased the product abroad, 3% (n = 22) from a herbalist, 2% (n = 14) from a market, and no respondents used a human licensed medical product purchased from a pharmacy. Twenty-two percent (n = 165) of the respondents additionally indicated a free-text option and indicated purchase through private traders (n = 45), retailers (n = 40), veterinarians (n = 20), alternative therapists (n = 20), and other (n = 39). Of the 20 respondents purchasing cannabinoid products through veterinarians, indications for use were: pain management (n = 12), behavioural issue (n = 6), cancer (n = 4), appetite stimulation (n = 1), kidney disease (n = 1), and allergy (n = 1).

## Use of other types of herbal remedies

In total, 30% (n = 609) of all respondents had used other herbal remedies. Of these, 44% (n = 265) had also used cannabinoids. The herbal remedies most often reported were aloe vera, Bach[®] Original Flower Remedies, Kalm (ScanVet Animal Health A/S) (l-tryptophan, milk protein, vitamin B and l-theanine), fish oil, homoeopathy, green-lipped mussels, chondroitin, hyaluronic acid, glucosamine, pro- and prebiotics, psyllium, valerian root, and yeast extracts. A complete overview of used herbal remedies is listed in S2 Appendix.

## Discussion

The study explored the use and perceived efficacy of cannabinoid treatment in dogs in Denmark. The results indicate that cannabinoids are used for a variety of reasons, and most are purchased online. The majority of respondents perceived that the treatment improved their dogs' medical condition or well-being. Despite the lack of licensed cannabinoid-containing products for pets, and even though it is illegal to sell or prescribe cannabinoids for animals in Denmark, this survey of over 2,000 dog owners found that 38% of the respondents had used a cannabinoid product. Three prior surveys in the US and Canada have explored the use and owner-perceived efficacy of cannabinoids among dog owners with sample sizes of 106, 632,

and 1,068 [36–38]. The proportion of dog owners using cannabinoids was higher in the previous US and Canadian studies (79.8% and 78.3%) compared to the current Danish study [36, 38]. The difference may be due to study selection bias or the difference in the availability of cannabinoid products in North America and Denmark. In Denmark, the recreational use of cannabis is illegal, and human medical cannabis prescription is in its infancy, which may affect owner's awareness and willingness to use cannabinoids in their dogs. This assumption is supported by a Slovenian study which found a slight association between owners' personal experience with cannabinoid and their use in pets [39].

The most often stated indications for cannabinoid treatment were pain management, behavioural issues, and allergy. Similar results were reported in the North American surveys, where pain management, anxiety, and inflammatory disease were the most common indications [36, 38]. Besides these indications, the respondents reported a very large range of indications for which they used cannabinoids.

The majority of dog owners perceived a positive effect of the cannabinoid treatment for various indications. This could be due to a placebo effect or selection bias, or it could reflect a real therapeutic effect or improved quality of life for the treated dogs. However, there are no veterinary clinical trials to support or refute these findings for indications other than osteoarthritis-related pain, noise-induced anxiety, aggression, voluntary activity, canine atopic dermatitis, and seizures [16, 19–24, 26–28, 30, 31]. The indication with the highest owner-perceived efficacy was pain management where 77% experienced a positive effect with the majority reporting an obvious and good effect. The most common cannabinoid used in this group was CBD. Support for beneficial effects of CBD in the management of osteoarthritis-related pain in dogs has been reported in three double-blinded, randomized clinical trials [16, 20, 24]. Gamble et al. (2018) performed a randomized, placebo-controlled, cross-over, and double-blinded study with administration of 2 mg/kg CBD every 12 hours in 16 dogs. At two and four weeks into treatment, the Canine Brief Pain Inventory (CBPI) and a Hudson activity score showed a decrease in pain and an increase in activity compared to baseline ($p < 0.01$). The subjective veterinary clinical assessment in the study also showed a decrease in pain scores from baseline ($p < 0.02$) [16]. Similar findings were reported by Verrico et al. (2020) who investigated the effect of two CBD doses and liposomal CBD in 20 dogs with osteoarthritis-related pain. They found a decrease in pain estimated by the Helsinki Chronic Pain Index (HCPI) ($p < 0.01$) in dogs treated with either 1.2 mg/kg/day CBD or 20 mg/day liposomal CBD [20]. Only one double-blinded, cross-over, and placebo-controlled study has used objective evaluation methods in addition to the subjective Clinical Metrology Instruments (CMI) in the investigation of the effect of 2.5 mg/kg CBD every 12 hours in osteoarthritis-related pain. Even though they found similar results in the subjective measurements (Liverpool Osteoarthritis in dogs and CBPI) as in the two previous studies, the objective measurements from accelerometry and objective gait analysis with pressure-sensitive walkway did not identify a significant difference between CBD and placebo groups after 6 weeks of treatment [24]. The disparity between the results from owner-driven subjective assessment tools (CMIs) and the objective assessment methods could indicate that the observations are reflections of CBD's effect on other factors influencing the quality of life (better quality of sleep, reduced anxiety, better ability to cope with pain) than a true analgesic effect. Recent systematic reviews and meta-analyses' have been conducted in the human field investigating the efficacy of cannabinoid treatment in different pain phenotypes. When evaluating the cannabinoid efficacy in chronic non-cancer pain patients, moderate quality of evidence for a small beneficial effect was reported [40], whilst when used in adult cancer-related pain patients the conclusion was that the effect of cannabinoids in addition to opioid treatment did not reduce pain in these patient phenotypes [41]. In our study, it was not possible to recognize which pain phenotypes (inflammatory pain, neuropathic pain, cancer-

induced pain, chronic pain, acute pain) the owners were treating, but if dogs experience the same pain phenotypes as is described in humans, there might be certain subgroups of dogs in which cannabinoids are more effective than others. In contrast to the veterinary studies the human studies are based on cannabinoid products containing both CBD and THC, which could affect the outcome and efficacy of pain management. The second largest indication category was behavioural issues which included but was not restricted to treatment of anxiety, stress, firework- and thunder phobia. The owner-perceived effect of CBD treatment was high (67%). These findings are not supported by current studies investigating behavioural modification in dogs after CBD treatment. One study investigated the effect of 1.25 mg/kg CBD daily on aggression and stereotypic behaviour and did not recognize a difference between the treatment group and placebo group (p = 0.078) [23]. A second study aimed to investigate the effect of 0.7 mg/kg CBD every 12 hours on firework phobia and did not register any anxiolytic effect (plasma cortisol levels, pulse, anxiety-related behaviour) after the CBD treatment [22]. Interestingly the category with the highest percentage of owners not observing an effect was in the seizure group where 12% reported no effect. McGrath et al. (2019) published the first clinical trial using 2.5 mg/kg CBD every 12 hours as adjunctive therapy in epileptic dogs and reported a reduction in seizure frequency in the treatment group compared to the placebo group (p <0.01). Supporting these findings is a study by Garcia et al. (2022) who reported a 50% reduction in epileptic activity in six out of 14 dogs when treated with 2 mg/kg CBD compared to no reduction in epileptic activity in the placebo group (p = 0.02) [31]. The relatively high percentage of perceived lack of effect compared to other indications in our study could reflect several causes. The seizure group had few individuals compared to the pain management and behavioural issue group as well as seizure activity can be a very obvious clinical sign that is difficult to neglect or overlook by the owner. Clinical signs in the other categories can be relatively difficult to assess for owners and may even be overlooked and could thereby have been perceived as treatment effects by the owner.

One surprising finding was that 20 respondents indicated that they had purchased cannabinoid products through their veterinarian. The veterinary market in Denmark is highly regulated and it is illegal for a veterinarian to prescribe cannabinoids. The indications for use of the veterinarian-prescribed cannabinoids were pain management, behavioural issues, cancer, appetite stimulation, kidney disease and allergy. Even though many in vitro and in vivo studies have shown promising tendencies for treatment of all of the above-mentioned indications [42, 43], it is only the use in osteoarthritis-related pain management, canine atopic dermatitis and behavioural issue that is evidence-based and supported by canine clinical trials [16, 20, 22, 23, 28, 30].

Our study suggests that CBD drops/oil are the most popular type and formulation of cannabinoids and was used by 93% of the respondents confirming use of cannabinoids. It is not possible from the survey to identify the precise cannabinoid content or concentration in the used products. None of the respondents indicated using registered human medical cannabinoid products, which are the only products with controlled and documented cannabinoid content legally available in Denmark. A concern with the use of unauthorized cannabinoid products is the lack of content specification and batch analysis. The reported products in this study have the potential of not containing any cannabinoids. It has been documented that a discouraging number of veterinary products labelled as containing specific types and amounts of cannabinoids, had divergent content [34]. Furthermore, some respondents indicated using pure hemp seed oil which does not contain cannabinoids except if contaminated [44]. The uncertainty of content in the cannabinoid products is challenging for both dosing recommendations and effect evaluation, especially because the appropriate therapeutic dose could vary for different medical conditions. For seizure control in dogs, there is an indication that plasma

concentration levels correlate with the effect [21], just as it has been shown that low-dose CBD (0.5 mg/kg/day) did not have the same positive effect on pain scores as higher CBD (1.2 mg/kg/day) concentrations [20]. Besides the challenges in dose recommendations and effect evaluation when using products with either unknown or varying cannabinoid content, it is especially critical regarding the THC concentration. Dogs are more sensitive to the psychoactive properties of THC compared to humans, possibly due to larger quantities of CB1 receptors in the brain, which can lead to high morbidity rates when exposed to THC [45]. In our study, very few reported intentionally using products primarily containing THC (4%).

Our study found that respondents living in Greater Copenhagen (Capital area) were less likely to supplement their dogs with cannabinoids. The reason for this is not clear, but it could be speculated to be linked to the higher educational level of the general population in this area, making these owners more adherent to evidence-based therapies, and less motivated for "alternative treatments". This statement is however contradicted in the Slovenian study which did not find owner educational status as a predictor of treating pet animals with cannabinoids [39].

The authors would like to address some of the limitations of this study. First, it should be emphasized that this survey of owner-perceived effects should not be seen as a validation of the efficacy of any of the mentioned cannabinoid formulations or types. The study was designed as a short simple questionnaire for distribution via social media to secure as many respondents as possible. The potential bias with this type of self-selected convenience sampling is that dog owners with very strong opinions for or against cannabinoid use could be more motivated to participate (selection bias) and caution should be taken in concluding that this is a true representation of the general population. Secondly, the recognition of treatment effect is based on owner observations with a lack of standardized efficacy evaluation tools and without placebo or control groups (detection bias), resulting in a risk of reporting high numbers of both placebo and nocebo effects. Furthermore, the survey does not consider the owner's capability to evaluate medical conditions and responses. Lastly, the treatments reported are not standardized or even documented regarding cannabinoid profile, content, or dosage.

## Conclusion and perspectives

Danish dog owners administer cannabinoids for a broad range of different medical and behavioural conditions, despite no legal products being available on the Danish market. The majority of the owners perceive a very good or good effect of the cannabinoid treatment, especially when used for pain management. Most owners purchase the cannabinoid product online and most often use CBD drops/oil.

While caution must be exercised in interpreting the results, this study supports the need for more evidence-based knowledge within the companion animal field of cannabinoids, as some dog owners self-prescribe cannabinoids for a wide variety of medical conditions in their dogs. Pet owners do request information on cannabinoid products and currently, it is challenging for veterinarians to provide evidence-based information and dosing recommendations. Pet owners are to a high degree left to seek information and recommendations from commercial websites, family, friends, or anecdotal sources, which may not be factual or impartial.

As the majority of studies have focused on osteoarthritis-related pain, it is possible that cannabinoids may have a therapeutic rationale for additional veterinary indications or health-related conditions, not yet explored. Based on the findings of this study the authors suggest conductance of larger double-blinded, randomized and controlled clinical and dose-escalating trials, especially in areas of different pain phenotypes, behavioural modulation, and allergy. Future studies should be conducted using quality-controlled products with defined and

documented cannabinoid content, profile, and concentrations and importantly should include objective assessment methods.

## Supporting information

**S1 Appendix. Survey questionnaire.** The questionnaire has for this publication been translated from the original survey language (Danish).
(PDF)

**S2 Appendix. Overview of reported use of herbal remedies by respondents.** The list has for this publication been translated from the original survey language (Danish).
(PDF)

**S1 File. Dataset.**
(XLSX)

## Acknowledgments

The authors would like to acknowledge and express their sincere thanks to Assistant Professor Anna Mueller, DVM, Ph.D. for input and management of the recruitment advert on the University Hospital for Companion Animals' Facebook page enabling the high respondent numbers, to the Danish Kennel Klub and their magazine "Hunden", as well as our veterinary colleagues who distributed the online survey and not least to all the Danish dog owners who showed interest and contributed to the study. We would also like to thank the Section of Biostatistics, Statistical Advisory Service, University of Copenhagen for their assistance with the statistical analysis.

## Author Contributions

**Conceptualization:** Maja Louise Arendt.

**Data curation:** Pernille Holst.

**Formal analysis:** Pernille Holst.

**Investigation:** Pernille Holst.

**Methodology:** Pernille Holst.

**Project administration:** Pernille Holst.

**Supervision:** Annemarie Thuri Kristensen, Maja Louise Arendt.

**Writing – original draft:** Pernille Holst, Annemarie Thuri Kristensen, Maja Louise Arendt.

**Writing – review & editing:** Pernille Holst, Annemarie Thuri Kristensen, Maja Louise Arendt.

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
