## [Decision Letter · Decision Letter 0]

20 Oct 2023

PONE-D-23-05678Danish dog owners’ use and the perceived effect of unlicensed cannabis products in dogsPLOS ONE

Dear Dr. Holst,

Thank you for submitting your manuscript to PLOS ONE. After careful consideration, we feel that it has merit but does not fully meet PLOS ONE’s publication criteria as it currently stands. Therefore, we invite you to submit a revised version of the manuscript that addresses the points raised during the review process.

Thank you for submitting this interesting manuscript I look forward to receiving your revisions. ==============================

We look forward to receiving your revised manuscript.

Kind regards,

Lucy J Troup, Ph.D

Academic Editor

PLOS ONE

Journal Requirements:

Reviewers' comments:

Reviewer's Responses to Questions

**Comments to the Author**

1. Is the manuscript technically sound, and do the data support the conclusions?

Reviewer #1: Yes

Reviewer #2: Yes

2. Has the statistical analysis been performed appropriately and rigorously? 

Reviewer #1: Yes

Reviewer #2: Yes

3. Have the authors made all data underlying the findings in their manuscript fully available?

Reviewer #1: Yes

Reviewer #2: Yes

4. Is the manuscript presented in an intelligible fashion and written in standard English?

Reviewer #1: Yes

Reviewer #2: Yes

5. Review Comments to the Author

Reviewer #1: First it should be noted I am not well versed in qualitative studies.

Research regarding the relationship between dogs and cannabinoids is in its infancy. Studies which aim to understand this relationship are extremely important. This study does a good job at beginning to highlight how and why cannabinoids are used on dogs in Denmark, and acts as an introduction to guide further research. Authors did an excellent job detailing the limitations of this study. Their inferences were conservative, but present very useful information for researchers to use moving forward. The previous reviewers have done a good job ruling out the majority of issues this manuscript had, and my comments are quite minor. The following are suggestions I have to finalize the manuscript before publishing.

Main comments

Throughout the manuscript there are issues with grammar. Although this has been addressed by the other reviewers, the remaining mistakes still take away from the message the authors want to send. It's a good idea to put the manuscript through "Grammarly", as it will likely catch the remaining grammatical errors this manuscript has (https://www.grammarly.com/). Be consistent with spacing between paragraphs and indentations at the beginning of paragraphs.

The authors state that one of their objectives it to highlight where future research could be focused. This is study is offers a unique perspective on future studies that hasn't been seen in equivalent studies performed elsewhere. As qualitative studies are important for hypothesis generation, it would be nice if the authors expanded on this (in either the discussion or conclusion) to specifically highlight what studies they would recommend.

Minor comments

Line 45-Space after registered.

Paragraph of line 55- Id recommend concluding this paragraph with the point of stating cannabis products are mislabeled or contaminated. I don't follow the relevance. Or maybe change it to highlight the dangers of self-prescribing.

Line 69- Replace "on till" with "until".

Line 82- Remove comma after "research"

Line 227-Table 3 isn't completely visible.

Line 239-Make it clear exactly what and why these tests are being performed, i.e., your goal.

Line 293-Be consistent with spacing between values and unites of measurement. This occurs throughout the manuscript.

Reviewer #2: With much interest I have read and reviewed the manuscript titled: ‘Danish dog owners’ use and the perceived effect of unlicensed cannabis products in dogs’. The researchers studied unlicensed cannabinoid use in Danish dogs, by questioning dog owners on usage, including reasons and perceived effectiveness.

I would recommend the study to be accepted for submission by PloSOne and feel the study is relevant as interest among dog owners seems increasing, regarding cannabis products as possibly relevant for instance as pain treatment and to modulate behaviour; yet presently studies on the topic are few. This study indicates how without a veterinary licence, cannabinoid products are opted for by Danish dog owners, how the products are bought, for which purposes and how owners perceive their effectiveness.

The study method is based on surveying a convenience sample, recruited via social media, with 38% of 2,002 respondents indicating use of cannabis products for the dog.

Minor revisions

- May I suggest to write 2,002 (and 2,091, 1,044, etc.) as to assist the reader in recognising it as a number?

- Line 50, ‘On the other hand’ -> perhaps another start of the sentence positions it more optimally? I was also wondering if the three surveys that are mentioned after this sentence, are not better addressed elsewhere? (Or alternatively the whole text from line 50 – 55 may be better positioned elsewhere? Note that line 77 mentions ‘anecdotal’ also. Perhaps the text may fit here, leading up to the study aim?)

- Line 70/71, although ‘physicians’ will be sufficient indication for native speakers, perhaps adding ‘human’ somewhere may aid non-native speakers in understanding rapidly that this text is about indications for human conditions/human patients?

- ‘Feedstuff’ (line 56/74, I am unsure here, but is this terminology not normally used for livestock more so than dogs (or other companion animals)? It may be that you intend to indicate livestock feedstuff?

- Would it be of relevance to (eg. near line 58) briefly explain that many cannabinoids from Cannabis sativa exist and that often cannabidiol (CBD) and 19-tetrahydrocannabinol (THC) are present in products in larger amounts ant that the latter is deemed toxic to dogs?

- Line 83: is ‘legality’ the optimal word choice?

- With regard to Methods, would you not want to indicate earlier on (than line 13) that the questionnaire is available as an Appendix?

- Line 114: ‘this section’ -> perhaps describe ‘this’, so which section, e.g. the section on cannabinoid product use?

- Line 119: ‘for which’ i.s.o. ‘which’?

- Line 158, did the questionnaires report?

- Line 160, is it ‘decision to treat’ or e.g. ‘reporting of treating the dog with…’?

- Line 162 ‘pet dogs’, but before this ‘dogs’. (Also in line 163.)

- Line 164: note that there is a discrepancy in # of decimal places between OR and CI.

- Line 173: is ‘positive respondents’ the optimal word choice? (Also in line 350).

- Line 175-177: ‘The evaluation of 36 open-ended text boxes provided by the respondents showed that some responses should have been included in the predefined categories.’ -> you may want to address in the Methods section how you evaluated the open-ended text data. This may also allow you to write more ‘result style’ in this section after line 175.

- When reading ‘Seizure control’ – line 188, I wondered how this related to ‘neurological disease including epilepsy’ – line 193. The explanation is found in line 201 onwards. You may want to (after providing detail in the Methods section, see previous comment) simply provide the results here after correctly categorizing data from open-ended text data. You can list how you categorized items in an Appendix. The benefit may be that results become graspable more easily for your readers. Table 2 is already valuable to do so, but you may reconsider positioning and writing as to also end this paragraph a little more strongly?

- Table 3 was not completely in view in the pdf.

- Line 229-230 ‘reported the highest degree of perceived effect’ -> you may want to check if this sentence reflects correctly what you aim to state here; is it the highest degree or e.g. the highest percentage of….

- Line 229-241 is interesting to read, but as a thought: The table presents the percentage and numbers per scale item, would it create a more graspable text if you’d mention here the binary effect versus no effect for each use purpose (indication) in this text part? This would perhaps facilitate a comparison between use purposes?

- Line 250: ‘Twenty-two percentage’ -> ‘percent’ or ‘a percentage of …’, I think?

- Line 258/259: are Kalm® and Bach® brand names? Then you may need to indicate that somehow?

- Line 266/297: ‘Despite the lack of licensed cannabinoid-containing products for pets, and ==despite the fact== that it is illegal to sell or prescribe (…).

- Line 275: ‘This == e.g. assumption/suggestion== is’.

- Line 278: ‘The most often stated indications for the cannabinoid treatment was pain management, (…)’ -> ‘were’ i.s.o. ‘was’?

- Line 317/318: ‘but it is likely that dogs also experience certain pain entities where cannabinoids are potentially more effective’ -> note the difference from pain type (terminology in previous sentences) and ‘pain entities’ and is it ‘likely’ (and if so, can you refence that?) or would you want to word the sentence a little more carefully?

- Line 326/327: font difference and ‘they’ may not be the optimal way to reflect back on ‘second study’.

- Line 337-340: Text ‘Furthermore (…) overlooked’: While understanding what you intend to indicate here, you may want to rewrite this text a little (and add some references?) as it may not be clear why to your reader why more obvious signs are more or less related to placebo effect? Not also that the final sentence part seem to miss a word or two (eg. ==may==even== be== overlooked).

- Line 343: indications? (plural).

- Well done for mentioning the study limitations clearly at the end of the discussion section. I was wondering about a possible influence of some other aspects, particularly if the (clearly described) choice of testing for associations only with the ‘1 treatment indication’ could have led to additional bias. (As for instance a very ‘contented’ person providing of cannabinoid products to a dog, may use more broadly than a less ‘contented’ person? However, this does not seem to regard a very large part of your dataset.

- Line 389: ‘medical’, does this include ‘behavioural’?

- Line 390: ‘are’=> ‘being’?

6. PLOS authors have the option to publish the peer review history of their article (what does this mean?). If published, this will include your full peer review and any attached files.

Reviewer #1: No

Reviewer #2: No

---

## [Author Response · Author response to Decision Letter 0]

4 Dec 2023

Dear Editors,

The authors would like to thank the reviewers and section editor for their careful reading and thoughtful comments that helped improve the quality and clarity of the manuscript. We especially want to thank the reviewers for the time and effort taken to review our study, and for their insightful and encouraging comments. 

We have addressed all the comments in the revised manuscript and have hopefully satisfied the raised concerns. The detail of the changes made is given below.

Response to Editor

## Thank you for bringing this to our attention. The style requirements have been reassessed and corrected.

## One reference has been added to the revised manuscript: Weiblen GD, Wenger JP, Craft KJ, ElSohly MA, Mehmedic Z, Treiber EL, et al. Gene duplication and divergence affecting drug content in Cannabis sativa. New Phytologist. 2015;208(4):1241–50. 

Response to Reviewer #1

First it should be noted I am not well versed in qualitative studies.

Research regarding the relationship between dogs and cannabinoids is in its infancy. Studies which aim to understand this relationship are extremely important. This study does a good job at beginning to highlight how and why cannabinoids are used on dogs in Denmark, and acts as an introduction to guide further research. Authors did an excellent job detailing the limitations of this study. Their inferences were conservative, but present very useful information for researchers to use moving forward. The previous reviewers have done a good job ruling out the majority of issues this manuscript had, and my comments are quite minor. The following are suggestions I have to finalize the manuscript before publishing.

## Thank you for your very generous and encouraging comments. We are very appreciative of your valuable input. To the best of our ability, we have revised the manuscript. The detail of the changes made is given below.

Throughout the manuscript there are issues with grammar. Although this has been addressed by the other reviewers, the remaining mistakes still take away from the message the authors want to send. It's a good idea to put the manuscript through "Grammarly", as it will likely catch the remaining grammatical errors this manuscript has (https://www.grammarly.com/). Be consistent with spacing between paragraphs and indentations at the beginning of paragraphs.

## Thank you very much for the grammatical, spelling, and structural edits throughout the manuscript and the recommendation of "Grammarly". The manuscript has been revised to the best of our ability.

The authors state that one of their objectives it to highlight where future research could be focused. This is study is offers a unique perspective on future studies that hasn't been seen in equivalent studies performed elsewhere. As qualitative studies are important for hypothesis generation, it would be nice if the authors expanded on this (in either the discussion or conclusion) to specifically highlight what studies they would recommend.

## We appreciate the comment. The revised manuscript has in addition to comments already made in the original manuscript’s Conclusion and Perspectives, been added text on future perspectives where we specify study details, (last paragraph in Conclusion and Perspectives).

Line 45- Space after registered.

## Thank you, space inserted.

Paragraph of line 55- Id recommend concluding this paragraph with the point of stating cannabis products are mislabeled or contaminated. I don't follow the relevance. Or maybe change it to highlight the dangers of self-prescribing.

## Thank you for highlighting this obvious point. We have added a concluding paragraph stating the concerns and potential health risks there are when owners self-prescribe these unregulated products.

Line 69- Replace "on till" with "until".

## Thank you for the spelling correction. Corrected in text.

Line 82- Remove comma after "research"

## Thank you for the grammatical edit. Corrected in text.

Line 227-Table 3 isn't completely visible.

## We are very sorry for the misunderstanding and for your inconvenience. We have contacted the publishing editor and have added an additional file labelled “Other_Table 3” to our submission of the revised paper. Please find the correspondence below.

Dear Editor

Thank you very much for the uplifting and positive response to our manuscript. We very much appreciate the reviewer’s inputs and we are addressing them at the moment.

I am however in doubt of one of the reviewer comments and I think I probably have misunderstood the guideline. I hope you can guide me on how to correct the issue.

We have a large table that extends beyond the margins and as I understood the guideline I should not try and fit it into the margins. I have most likely misunderstood the guidelines, as both reviewers have commented on it. Could you clarify how I should correct it in order to meet the reviewer’s comments.

From guideline:

Size: Tables do not have strict width and height requirements. Do not split your table or otherwise try to make the table appear within the manuscript margins if it does not fit on one page. In Word, tables that run off of the manuscript page can be seen using Draft View. In the PDF version of the published article, very wide tables may be printed sideways, and long tables may span more than one page. 

Thank you very much, Pernille

Dear Dr. Holst,

Thank you for reaching out with your query.

You're correct, we do ask that you don't alter your tables to fit. This is in case of acceptance, the production team will use the tables as they're shown in the manuscript to put together the publishable version. When viewing your manuscript in Word using the draft view, the full Table 3 is completely visible, However, in the PDF file which is provided the the reviewers, the table is cut off.

I'd recommend when submitting your revised paper to keep the table in the manuscript as it is, but to also upload the table as a separate "Other" file. You can reference the separate file in your Response to Reviewers so the reviewers know this is the same Table 3 that's included in the manuscript and are able to examine the full table.

My apologies for the misunderstanding. Please let me know if you have any further questions or if you have concerns about this suggested solution.

Kind regards, Teresa Diviacchi, Publishing Editor

Line 239- Make it clear exactly what and why these tests are being performed, i.e., your goal.

## Thank you for the very relevant concern. We have rephrased the paragraph to state the intention of the performed tests more clearly. The paragraph reads; As differences in the owner perceived effect between the indication groups could be a reflection of a true difference in response to treatment of different disease states a Kruskal-Wallis test was performed and provided weak evidence of a difference χ 2(9, N = 516) = 19.808, p < 0.019, Ε2 = 0.40. A multiple comparison post hoc test with a Dunn's test did however not show any significant differences in perceived effect between the indication groups (p >0.05, adjusted using the Bonferroni correction).

Line 293- Be consistent with spacing between values and unites of measurement. This occurs throughout the manuscript.

## Thank you for the structural correction. The manuscript has been checked for spacing errors. 

Response to Reviewer #2

With much interest I have read and reviewed the manuscript titled: ‘Danish dog owners’ use and the perceived effect of unlicensed cannabis products in dogs’. The researchers studied unlicensed cannabinoid use in Danish dogs, by questioning dog owners on usage, including reasons and perceived effectiveness.

I would recommend the study to be accepted for submission by PloSOne and feel the study is relevant as interest among dog owners seems increasing, regarding cannabis products as possibly relevant for instance as pain treatment and to modulate behaviour; yet presently studies on the topic are few. This study indicates how without a veterinary licence, cannabinoid products are opted for by Danish dog owners, how the products are bought, for which purposes and how owners perceive their effectiveness. The study method is based on surveying a convenience sample, recruited via social media, with 38% of 2,002 respondents indicating use of cannabis products for the dog.

## Thank you for your interest in our study and your valuable input. We are very appreciative and have to the best of our ability revised the manuscript. The detail of the changes made is given below.

- May I suggest to write 2,002 (and 2,091, 1,044, etc.) as to assist the reader in recognising it as a number?

## Thank you for this very valid point – text edited.

- Line 50, ‘On the other hand’ -> perhaps another start of the sentence positions it more optimally? I was also wondering if the three surveys that are mentioned after this sentence, are not better addressed elsewhere? (Or alternatively the whole text from line 50 – 55 may be better positioned elsewhere? Note that line 77 mentions ‘anecdotal’ also. Perhaps the text may fit here, leading up to the study aim?)

## Thank you for the suggestion. We have revised the text and have rearranged the paragraph.

- Line 70/71, although 'physicians' will be sufficient indication for native speakers, perhaps adding 'human' somewhere may aid non-native speakers in understanding rapidly that this text is about indications for human conditions/human patients?

## Thank you, we indeed agree with your reflection – text adjusted.

- 'Feedstuff' (line 56/74, I am unsure here, but is this terminology not normally used for livestock more so than dogs (or other companion animals)? It may be that you intend to indicate livestock feedstuff?

## Thank you for the insightful comment and we believe your understanding of the word feedstuff is correct. We apologize for the misuse of the word. In the Danish legislation all non-medical cannabinoid compounds that are produced by extraction are categorized as “feedstuff” (Danish: foderstof) disregarding what species they are intended. More appropriate would probably be the term “feed additive”, which also is the wording used in the EU legislation. Feedstuff has been substituted with feed additive in the text.

- Would it be of relevance to (eg. near line 58) briefly explain that many cannabinoids from Cannabis sativa exist and that often cannabidiol (CBD) and 19-tetrahydrocannabinol (THC) are present in products in larger amounts ant that the latter is deemed toxic to dogs?

## Thank you for this comment, we agree with this suggestion. We have now extended the Introduction and included a section clarifying cannabinoid content and potential risk of intoxication (Introduction, first and second paragraph).

- Line 83: is 'legality' the optimal word choice?

## Thank you for bringing this to our attention – the sentence is reformulated and reads: Given the lack of legal veterinary cannabinoid products, an anonymous survey was deemed the most appropriate method to obtain this information.

- With regard to Methods, would you not want to indicate earlier on (than line 13) that the questionnaire is available as an Appendix?

## We agree with the comment and have moved the Appendix reference to the third paragraph in the “Survey” section.

- Line 114: 'this section' -> perhaps describe 'this', so which section, e.g. the section on cannabinoid product use?

## Thank you for pointing this out. The text has been revised and read: The respondents were questioned about their use or past use of cannabinoid products for their dogs. Those who confirmed usage were directed to a subsequent section containing more detailed inquiries….

- Line 119: 'for which' i.s.o. ‘which’?

## Thank you – text corrected.

- Line 158, did the questionnaires report?

## Thank you for this obvious point – text revised and reads: Of the 2,002 completed questionnaires, 752 (38%) respondents reported having used at least one cannabinoid product for their dog, while 1,250 (62%) had never used such products.

- Line 160, is it 'decision to treat' or e.g. 'reporting of treating the dog with…'?

## We agree with the comment and have revised the text which reads: A chi-square test was used to assess the association of living area, with the reporting of treating with cannabinoids, and a significant association was found (X2 (7, N=2,002) = 16.61, p = 0.020).

- Line 162 'pet dogs', but before this 'dogs’. (Also in line 163.)

## Thank you for the comments, the text has been corrected for this inconsistency.

- Line 164: note that there is a discrepancy in # of decimal places between OR and CI.

## Thank you for bringing this to our attention. The decimals have been uniformed and read: Binominal logistic regression revealed that living in greater Copenhagen was negatively associated with cannabinoid use in dogs, with the respondents having an odds ratio of 0.69 related to using cannabinoids (P = 0.04, OR 0.69, 95% CI = 0.48-0.98). 

- Line 173: is 'positive respondents' the optimal word choice? (Also in line 350).

## Thank you for the input. We agree and have changed the wording in the revised manuscript to:

Original line 173: Of all the respondents confirming the use of cannabinoids, 113 (15%) had used more than one cannabinoid formulation or type, with the majority having tried two different formulations or types. Original line 350: Our study suggests that CBD drops/oil are the most popular type and formulation of cannabinoids and were used by 93% of the respondents confirming the use of cannabinoids.

- Line 175-177: ‘The evaluation of 36 open-ended text boxes provided by the respondents showed that some responses should have been included in the predefined categories.’ -> you may want to address in the Methods section how you evaluated the open-ended text data. This may also allow you to write more ‘result style’ in this section after line 175.

## Thank you for the suggestion. The Material and Methods section has been revised and reads: The open-ended textboxes were checked for answers which should have been included in the predefined categories and the data was adjusted so they were included in the appropriate predefined categories. The Results section has been adjusted to: The evaluation of 36 open-ended text box answers showed that 15 of the responses should have been included in the predefined categories. When adding these (six CBD oil, two CBD cream, and seven CBD capsules) to the appropriate categories, the overall frequencies were not affected.

- When reading 'Seizure control' – line 188, I wondered how this related to 'neurological disease including epilepsy' – line 193. The explanation is found in line 201 onwards. You may want to (after providing detail in the Methods section, see previous comment) simply provide the results here after correctly categorizing data from open-ended text data. You can list how you categorized items in an Appendix. The benefit may be that results become graspable more easily for your readers. Table 2 is already valuable to do so, but you may reconsider positioning and writing as to also end this paragraph a little more strongly?

##Agreed, we have created confusion here by including “epilepsy” in the “neurological disease” free text category. The text has been revised.

- Table 3 was not completely in view in the pdf.

## We are very sorry for the misunderstanding and for your inconvenience. We have contacted the publishing editor and have added an additional file labelled “Other_Table 3” to our submission of the revised paper. Please find the correspondence below.

Dear Editor

Thank you very much for the uplifting and positive response to our manuscript. We very much appreciate the reviewers' input, and we are addressing them at the moment.

I am however in doubt of one of the reviewer comments and I think I probably have misunderstood the guideline. I hope you can guide me how to correct the issue.

We have a large table that extends beyond the margins and as I understood the guideline I should not try and fit it into the margins. I have most likely misunderstood the guidelines, as both reviewers have commented on it. Could you clarify how I should correct it in order to meet the reviewer’s comments.

From guideline:

Size: Tables do not have strict width and height requirements. Do not split your table or otherwise try to make the table appear within the manuscript margins if it does not fit on one page. In Word, tables that run off of the manuscript page can be seen using Draft View. In the PDF version of the published article, very wide tables may be printed sideways, and long tables may span more than one page. 

Thank you very much, Pernille

Dear Dr. Holst,

Thank you for reaching out with your query.

You're correct, we do ask that you don't alter your tables to fit. This is in case of acceptance, the production team will use the tables as they're shown in the manuscript to put together the publishable version. When viewing your manuscript in Word using the draft view, the full Table 3 is completely visible, However, in the PDF file which is provided the the reviewers, the table is cut off.

I'd recommend when submitting your revised paper to keep the table in the manuscript as it is, but to also upload the table as a separate "Other" file. You can reference the separate file in your Response to Reviewers so the reviewers know this is the same Table 3 that's included in the manuscript and are able to examine the full table.

My apologies for the misunderstanding. Please let me know if you have any further questions or if you have concerns about this suggested solution.

Kind regards, Teresa Diviacchi, Publishing Editor

- Line 229-230 'reported the highest degree of perceived effect' -> you may want to check if this sentence reflects correctly what you aim to state here; is it the highest degree or e.g. the highest percentage of….

## Thank you for pointing this out to us. Degree has been changed to percentage.

- Line 229-241 is interesting to read, but as a thought: The table presents the percentage and numbers per scale item, would it create a more graspable text if you'd mention here the binary effect versus no effect for each use purpose (indication) in this text part? This would perhaps facilitate a comparison between use purposes?

##Thank you very much for this very relevant suggestion which we initially also considered by grouping the responses binary into "Effect" (including "Yes, very convincing and good effect" + "Yes, convincing with some effect" + "Possible, but not convincing effect") versus "No Effect" or alternatively “Effect” (including categories “Yes, very convincing and good effect” + “Yes, convincing with some effect”) versus “No Effect” (including “Possible, but not convincing effect” + “No Effect”).

In hindsight we have had doubts about how owners perceive the category “Possible, but not convincing effect” and we have had concerns about over-interpreting this category and have chosen not to include it in either effect or no effect. We have therefore chosen to keep the original text where only "Yes, very convincing and good effect" + "Yes, convincing with some effect" are considered an effect and summed in the text.

- Line 250: 'Twenty-two percentage' -> 'percent' or 'a percentage of …', I think?

## Thank you – spelling corrected.

- Line 258/259: are Kalm® and Bach® brand names? Then you may need to indicate that somehow?

## Thank you for this obvious point. The text has been revised and reads: The herbal remedies most often reported were aloe vera, Bach® Original Flower Remedies, Kalm (ScanVet Animal Health A/S) (l-tryptophan, milk protein, vitamin B and l-theanine), fish oil, homoeopathy, green-lipped mussels, chondroitin, hyaluronic acid, glucosamine, pro- and prebiotics, psyllium, valerian root, and yeast extracts.

- Line 266/297: 'Despite the lack of licensed cannabinoid-containing products for pets, and ==despite the fact== that it is illegal to sell or prescribe (…).

## Thank you for this comment, the text is revised.

- Line 275: 'This == e.g. assumption/suggestion== is’.

## Thank you for this comment, the text is revised and reads: This assumption is supported by a Slovenian study which found a slight association between owners’ personal experience with cannabinoid and their use in pets.

- Line 278: 'The most often stated indications for the cannabinoid treatment was pain management, (…)' -> 'were' i.s.o. ‘was’?

## Thank you – the text was corrected.

- Line 317/318: 'but it is likely that dogs also experience certain pain entities where cannabinoids are potentially more effective' -> note the difference from pain type (terminology in previous sentences) and 'pain entities' and is it 'likely' (and if so, can you refence that?) or would you want to word the sentence a little more carefully?

## Thank you for pointing this out and we do agree that the statement is too blunt. The text has been revised and reads: In our study, it was not possible to recognize which pain phenotypes (inflammatory pain, neuropathic pain, cancer-induced pain, chronic pain, acute pain) the owners were treating, but if dogs experience the same pain phenotypes as is described in humans, there might be certain subgroups of dogs in which cannabinoids are more effective than others. 

- Line 326/327: font difference and 'they' may not be the optimal way to reflect back on ‘second study’.

## Thank you – the text was corrected.

- Line 337-340: Text 'Furthermore (…) overlooked': While understanding what you intend to indicate here, you may want to rewrite this text a little (and add some references?) as it may not be clear why to your reader why more obvious signs are more or less related to placebo effect? Not also that the final sentence part seem to miss a word or two (eg. ==may==even== be== overlooked).

## Thank you for pointing this out. We agree that the text in the manuscript is misleading in the sense that the placebo effect is directly linked to the owners' ability to assess treatment response. We have revised the text in the manuscript.

- Line 343: indications? (plural).

## Thank you – the text was corrected.

- Well done for mentioning the study limitations clearly at the end of the discussion section. I was wondering about a possible influence of some other aspects, particularly if the (clearly described) choice of testing for associations only with the '1 treatment indication' could have led to additional bias. (As for instance a very 'contented' person providing of cannabinoid products to a dog, may use more broadly than a less 'contented' person? However, this does not seem to regard a very large part of your dataset.

## Thank you for your comment and acknowledging the biased dataset and our efforts to not over-interpretate the data. We fully agree that there is additional bias, especially regarding our selection of included data.

- Line 389: 'medical', does this include 'behavioural'?

## Agreed, the term medical is probably not comprehensive enough. Although some of the behavioural issues probably will be medical conditions, there are very likely conditions that would not be classified as a medical conditions but rather behavioural issues. We have included "behavioural conditions" in the sentence.

- Line 390: 'are'=> 'being'?

## Thank you. Text corrected.

Thank you once again for the possibility to re-submit.

With kind regards

Pernille Holst

---

## [Decision Letter · Decision Letter 1]

18 Dec 2023

Danish dog owners’ use and the perceived effect of unlicensed cannabis products in dogs

PONE-D-23-05678R1

Dear Dr. Holst,

We’re pleased to inform you that your manuscript has been judged scientifically suitable for publication and will be formally accepted for publication once it meets all outstanding technical requirements.

Kind regards,

Lucy J Troup, Ph.D

Academic Editor

PLOS ONE

Additional Editor Comments (optional):

Reviewers' comments:

Reviewer's Responses to Questions

**Comments to the Author**

1. If the authors have adequately addressed your comments raised in a previous round of review and you feel that this manuscript is now acceptable for publication, you may indicate that here to bypass the “Comments to the Author” section, enter your conflict of interest statement in the “Confidential to Editor” section, and submit your "Accept" recommendation.

Reviewer #1: All comments have been addressed

Reviewer #2: All comments have been addressed

2. Is the manuscript technically sound, and do the data support the conclusions?

Reviewer #1: Yes

Reviewer #2: Yes

3. Has the statistical analysis been performed appropriately and rigorously? 

Reviewer #1: Yes

Reviewer #2: Yes

4. Have the authors made all data underlying the findings in their manuscript fully available?

Reviewer #1: Yes

Reviewer #2: Yes

5. Is the manuscript presented in an intelligible fashion and written in standard English?

Reviewer #1: Yes

Reviewer #2: Yes

6. Review Comments to the Author

Reviewer #1: Thank you for addressing my comments.

There are still some grammatical errors. Put it through "Grammarly" again to catch them.

The use of spacing between paragraphs and indentations at the begining of paragraphs are still not consistent. Pick one style and stick with that style throughout the entire manuscript.

Quickly address those minor issues before publishing. I dont feel the manuscript needs to go through another round of revisions.

Good job and best of luck!

Reviewer #2: The authors have addressed all comments well and the improved manuscript will provide interesting information for its readers after publishing.

7. PLOS authors have the option to publish the peer review history of their article (what does this mean?). If published, this will include your full peer review and any attached files.

Reviewer #1: No

Reviewer #2: No

---

## [Editor Report · Acceptance letter]

4 Jan 2024

PONE-D-23-05678R1 

PLOS ONE

Dear Dr. Holst, 

I'm pleased to inform you that your manuscript has been deemed suitable for publication in PLOS ONE. Congratulations! Your manuscript is now being handed over to our production team.

Kind regards, 

on behalf of

Dr. Lucy J Troup 

Academic Editor

PLOS ONE